# ATTENTION-BASED GRAPH NEURAL NETWORK FOR SEMI-SUPERVISED LEARNING

## ABSTRACT

Recently popularized graph neural networks achieve the state-of-the-art accuracy on a number of standard benchmark datasets for graph-based semi-supervised learning, improving significantly over existing approaches. These architectures alternate between a propagation layer that aggregates the hidden states of the local neighborhood and a fully-connected layer. Perhaps surprisingly, we show that a linear model, that removes all the intermediate fully-connected layers, is still able to achieve a performance comparable to the state-of-the-art models. This significantly reduces the number of parameters, which is critical for semi-supervised learning where number of labeled examples are small. This in turn allows a room for designing more innovative propagation layers. Based on this insight, we propose a novel graph neural network that removes all the intermediate fully-connected layers, and replaces the propagation layers with attention mechanisms that respect the structure of the graph. The attention mechanism allows us to learn a dynamic and adaptive local summary of the neighborhood to achieve more accurate predictions. In a number of experiments on benchmark citation networks datasets, we demonstrate that our approach outperforms competing methods. By examining the attention weights among neighbors, we show that our model provides some interesting insights on how neighbors influence each other.

## 1 INTRODUCTION

One of the major bottlenecks in applying machine learning in practice is collecting sizable and reliable labeled data, essential for accurate predictions. One way to overcome the problem of limited labeled data is semi-supervised learning, using additional unlabeled data that might be freely available. In this paper, we are interested in a scenario when this additional unlabeled data is available in a form of a graph. The graph provides underlying pairwise relations among the data points, both labeled and unlabeled.

Of particular interest are those applications where the presence or absence of an edge between two data points is determined by nature, for instance as a result of human activities or natural relations. As a concrete example, consider a citation network. Each node in the graph is a published research paper, associated with a bag-of-words feature vector. An (directed) edge indicates a citation link. Presence of an edge indicates that the authors of a paper have consciously determined to refer to the other paper, and hence captures some underlying relation that might not be inferred from the bag-of-words feature vectors alone. Such external graph data are available in several applications of interest, such as classifying users connected via a social network, items and customers connected by purchase history, users and movies connected by viewing history, and entities in a knowledge graph connected by relationships. In this paper, we are interested in the setting where the graph is explicitly given and represents additional information not present in the feature vectors.

The goal of such *graph-based semi-supervised learning* problems is to classify the nodes in a graph using a small subset of labeled nodes and all the node features. There is a long line of literature on this topic since Blum & Chawla (2001) which seeks graph cuts that preserve the known labels and Zhu et al. (2003) which uses graph Laplacian to regularize the nearby nodes to have similar labels. However, Kipf & Welling (2016) recently demonstrated that the existing approaches can be significantly improved upon on a number of standard benchmark datasets, using an innovative neural network architecture on graph-based data known collectively as *graph neural networks*.

Inspired by this success, we seek to understand the reason behind the power of graph neural networks, to guide our design of a novel architecture for semi-supervised learning on graphs. To this end, we first found that a linear classifier of multinomial logistic regression achieves the accuracy comparable to the best known graph neural network. This linear classifier removes all intermediate non-linear activation layers, and only keeps the linear propagation function from neighbors in graph neural networks. This suggests the importance of aggregation information form the neighbors in the graph. This further motivates us to design a new way of aggregating neighborhood information through attention mechanism since, intuitively, neighbors might not be equally important. This proposed *attention-based graph neural network* captures this intuition and ($a$) greatly reduces the model complexity, with only a single scalar parameter at each intermediate layer; ($b$) discovers dynamically and adaptively which nodes are relevant to the target node for classification; and ($c$) improves upon state-of-the-art methods in terms of accuracy on standard benchmark datasets. Further, the learned attention strengths provide some form of *interpretability*. They provide insights on why a particular prediction is made on a target node and which neighbors are more relevant in making that decision.

## 2 RELATED WORK

Given a graph $G(V, E)$ with a set of $n$ nodes $V$ and a set of edges $E$, we let $X_i \in \mathbb{R}^{d_x}$ denote the feature vector at node $i$ and let $Y_i$ denote the true label. We use $Y_L$ to denote the labels that are revealed to us for a subset $L \subset V$. We let $X = [X_1, \ldots, X_n]$ denote all features, labeled and unlabeled.

Traditionally, semi-supervised learning using both labeled and un-labled data has been solved using two different approaches - Graph Laplacian based algorithms solving for locally consistent solutions (Zhou et al., 2004b) and Expectation Maximization based algorithms (Nigam et al., 2006) where true-labels of the unlabeled data points are considered as the latent variables of a generative model.

**Graph Laplacian regularization.** Based on the assumption that nearby nodes in a graph are more likely to have the same labels, the graph information has been used as explicit regularization:

$$\mathcal{L}(X, Y_L) \quad = \quad \mathcal{L}_{\text{label}}(X_L, Y_L) + \lambda \, \mathcal{L}_G(X) \, ,$$

where $\mathcal{L}_{\text{label}} = \sum_{i \in L} l(Y_i, f(X_i))$ is the standard supervised loss for some loss functions $l$ and $\mathcal{L}_G$ is the graph-based regularization, for example $\mathcal{L}_G = \sum_{(i,j) \in E} \|f(X_i) - f(X_j)\|^2$, which is called the graph Laplacian regularization. Earlier approaches are non-parametric and searches over all $f$ considering it as a look-up table. Most popular one is the Label Propagation (Zhu & Ghahramani, 2002) that forces the estimated labels to agree in the labeled instances and uses weighted graph Laplacian. This innovative formulation admits a closed form solution which makes it practically attractive with very low computationally cost. ManiReg (Belkin et al., 2006) replaces supervised loss with that of a support vector machine. ICA (Lu & Getoor, 2003) generalizes LP by allowing more general local updates. A more thorough survey on using non-neural network methods for semi-supervised learning can be found in (Chapelle et al., 2009).

More recent approaches are parametric, using deep neural networks. SemiEmb (Weston et al., 2012) was the first to use a deep neural network to model $f(x)$ and minimize the above loss. Planetoid (Yang et al., 2016) significantly improves upon the existing graph regularization approaches by replacing the regularization by another loss based on skip-grams (defined below). In a slightly different context, Buchnik & Cohen (2017) show that the accuracy of these approaches can be further improved by bootstrapping these models sequentially.

**Unsupervised node embedding for semi-supervised learning.** Several approaches have been proposed to embed the nodes in some latent Euclidean space using only the connectivity in graph $G$. Once the embedding is learned, standard supervised learning is applied on those embedded features to train a model. Inspired by the success of word2vec (Le & Mikolov, 2014), several approaches define "skip-grams" on graphs as the neighborhood (context) of a node on the graph and tries to maximize the posterior probability of observing those skip-grams. DeepWalk (Perozzi et al., 2014) and node2vec (Grover & Leskovec, 2016) use random walks as skip-grams, LINE (Tang et al., 2015) uses local proximities, LASAGNE (Faerman et al., 2017) uses the Personalized PageRank random walk. Graph2Gauss (A. Bojchevski, 2017) represents a node as a Gaussian distribution, and minimizes the divergence between connected pairs. Yang et al. (2017) provide a post-processing scheme

that takes any node embedding and attempts to improve it by by taking the weighted sum of the given embeddings with Personalized PageRank weights. The strength of these approaches is universality, as the node embedding does not depend on the particular task at hand (and in particular the features or the labels). However, as they do not use the node features and the training only happens after embedding, they cannot meet the performance of the state-of-the-art approaches (see DeepWalk in Table 2).

**Graph Neural Network (GNN).** Graph neural networks are extensions of neural networks to structured data encoded as a graph. Originally introduced as extensions of recurrent neural networks, GNNs apply recurrent layers to each node with additional *local averaging* layer (Gori et al., 2005; Scarselli et al., 2009). However, as the weights are shared across all nodes, GNNs can also be interpreted as extensions of convolutional neural networks on a 2D grid to general graphs. Typically, a message aggregation step followed by some neural network architecture is iteratively applied. The model parameters are trained on (semi-)supervised examples with labels. We give a typical example of a GNN in Section 3, but several diverse variations have been proposed in (Bruna et al., 2013; Duvenaud et al., 2015; Li et al., 2015; Henaff et al., 2015; Sukhbaatar et al., 2016; Bronstein et al., 2016; Defferrard et al., 2016; Dai et al., 2016; Atwood & Towsley, 2016; Niepert et al., 2016; Such et al., 2017; Hamilton et al., 2017; Schlichtkrull et al., 2017; Such et al., 2017). GNNs have been successfully applied in diverse applications such as molecular activation prediction (Gilmer et al., 2017), community detection (Bruna & Li, 2017), matrix completion (Berg et al., 2017), combinatorial optimization (Dai et al., 2017; Nowak et al., 2017), and detecting similar binary codes (Xu et al., 2017).

In particular, for the benchmark datasets that we consider in this paper, Kipf & Welling (2016) proposed a simple but powerful architecture called Graph Convolutional Network (GCN) that achieves the state-of-the-art accuracy. In the following section, $(a)$ we show that the performance of GCN can be met by a linear classifier; and $(b)$ use this insight to introduce novel graph neural networks that compare favourably against the state-of-the-art approaches on benchmark datasets.

## 3 DISSECTION OF GRAPH NEURAL NETWORK

In this section, we propose a novel Graph Neural Network (GNN) model which we call *Attention-based Graph Neural Network* (AGNN), and compare its performance to state-of-the-art models on benchmark citation networks in Section 5. We seek a model $Z = f(X, A) \in \mathbb{R}^{n \times d_y}$ that predicts at each node one of the $d_y$ classes. $Z_{ic}$ is the estimated probability that the label at node $i \in [n]$ is $c \in [d_y]$ given the features $X$ and the graph $A$. The data features $X \in \mathbb{R}^{n \times d_x}$ has at each row $d_x$ features for each node, and $A \in \{0, 1\}^{n \times n}$ is the adjacency matrix of $G$.

The forward pass in a typical GNN alternates between a *propagation layer* and a *single layer perceptron*. Let $t$ be the layer index. We use $H^{(t)} \in \mathbb{R}^{n \times d_h}$ to denote the current (hidden) states, with the $i$-th row $H_i^{(t)}$ as the $d_h$ dimensional hidden state of node $i$. A propagation layer with respect to a propagation matrix $P \in \mathbb{R}^{n \times n}$ is defined as

$$\tilde{H}^{(t)} = P H^{(t)} .$$

For example, the natural random walk[1] $P = D^{-1}A$ gives $\tilde{H}_i^{(t)} = (1/|N(i)|) \sum_{j \in N(i)} H_j^{(t)}$. The neighborhood of node $i$ is denoted by $N(i)$, and $D = \text{diag}(A\mathbb{1})$. This is a simple *local averaging* common in consensus or random walk based approaches. Typical propagation layer respects the adjacency pattern in $A$, performing a variation of such local averaging. GNNs encode the graph structure of $A$ into the model via this propagation layer, which can be also interpreted as performing a graph convolution operation as discussed in Kipf & Welling (2016). Next, a single layer perceptron is applied on each node separately and the weights $W^{(t)}$ are shared across all the nodes:

$$H^{(t+1)} = \sigma(\tilde{H}^{(t)} W^{(t)}) ,$$

where $W^{(t)} \in \mathbb{R}^{d_{h_{t+1}} \times d_{h_t}}$ is the weight matrix and $\sigma(\cdot)$ is an entry-wise activation function. This weight sharing reduces significantly the number of parameters to be trained, and encodes the invariance property of graph data, i.e. two nodes that are far apart but have the similar neighboring

---

[1] Random walk which moves from a node to one of its neighbors selected uniformly at random.

features and structures should be classified similarly. There are several extensions to this model as discussed in the previous section, but this standard graph neural network has proved powerful in several problems over graphs, e.g. (Bruna & Li, 2017; Berg et al., 2017; Dai et al., 2017).

**Graph Convolutional Network (GCN).** Kipf & Welling (2016) introduced a simple but powerful architecture, and achieved the state-of-the-art performance in benchmark citation networks (see Table 2). GCN is a special case of GNN which stacks two layers of specific propagation and perceptron:

$$
\begin{aligned}
H^{(1)} &= \text{ReLU}\big((PX)\,W^{(0)}\big)\,, \\
Z = f(X, A) &= \text{softmax}\big((PH^{(1)})\,W^{(1)}\big)\,,
\end{aligned}
\tag{1}
$$

with a choice of $P = \tilde{D}^{-1/2}\tilde{A}\tilde{D}^{-1/2}$, where $\tilde{A} = A + \mathbb{I}$, $\mathbb{I}$ is the identity matrix, $\tilde{D} = \text{diag}(\tilde{A}\mathbb{1})$ and $\mathbb{1}$ is the all-ones vector. $\text{ReLU}(a) = \max\{0, a\}$ is an entry-wise rectified linear activation function, and $\text{softmax}([a_1, \ldots, a_k]) = (1/Z)[\exp(a_1), \ldots, \exp(a_k)]$ with $Z = \sum_i \exp(a_i)$ is applied row-wise. Hence, the output is the predicted likelihoods on the $d_y$ dimensional probability simplex. The weights $W^{(0)}$ and $W^{(1)}$ are trained to minimize the cross-entropy loss over all labeled examples $L$:

$$
\mathcal{L} = -\sum_{i \in L}\sum_{c=1}^{d_y} Y_{ic} \ln Z_{ic}\,.
\tag{2}
$$

**Graph Linear Network (GLN).** To better understand GCN, we remove the intermediate nonlinear activation units from GCN, which gives Graph Linear Network defined as

$$
Z = f(X, A) = \text{softmax}\big((P^2 X)\,W^{(0)}\,W^{(1)}\big)\,,
\tag{3}
$$

with the same choice of $P = \tilde{D}^{-1/2}\tilde{A}\tilde{D}^{-1/2}$ as in GCN. The weights $W^{(0)}$ and $W^{(1)}$ have the same dimensions as GCN and are trained on a cross entropy loss in (2). The two propagation layers simply take (linear) local average of the raw features weighted by their degrees, and at the output layer a simple linear classifier (multinomial logistic regression) is applied. This allows us to separate the gain in the *linear* propagation layer and the *non-linear* perceptron layer.

Comparing the differences in performances in Table 2, we show that, perhaps surprisingly, GLN achieves an accuracy comparable to the that of the best GNN, and sometimes better. This suggests that, for citation networks, the strength of the general GNN architectures is in the propagation layer and not in the perceptron layer. On the other hand, the propagation layers are critical in achieving the desired performance, as is suggested in Table 2. There are significant gaps in accuracy for those approaches not using the graph, i.e. T-SVM, and also those that use the graph differently, such as Label Propagation (LP) and Planetoid. Based on this observation, we propose replacing the propagation layer of GLN with an attention mechanism and test it on the benchmark datasets.

## 4    ATTENTION-BASED GRAPH NEURAL NETWORK (AGNN).

The original propagation layer in GCN and several other graph neural networks such as (Defferrard et al., 2016; Atwood & Towsley, 2016; Monti et al., 2016; Such et al., 2017) use a static (does not change over the layers) and non-adaptive (does not take into account the states of the nodes) propagation, e.g. $P_{ij} = 1/\sqrt{|N(i)|\,|N(j)|}$. Such propagations are not able to capture which neighbor is more *relevant* to classifying a target node, which is critical in real data where not all edges imply the same types or strengths of relations.

We need novel *dynamic* and *adaptive* propagation layers, capable of capturing the relevance of different edges, which leads to more complex graph neural networks with more parameters. However, training such complex models is challenging in the semi-supervised setting, as the typical number of samples we have for each class is small; it is 20 in the standard benchmark dataset. This is evidenced in Table 2 where more complex graph neural network models by Verma et al. (2017), Monti et al. (2016), and Such et al. (2017) do not improve upon the simple GCN.

On the other hand, our experiments with GLN suggests that we can remove all the perceptron layers and focus only on improving the propagation layers. To this end, we introduce a novel Attention-based Graph Neural Network (AGNN). AGNN is simple; it only has a single scalar parameter $\beta^{(t)}$ at

each intermediate layer. AGNN captures relevance; the proposed *attention mechanism over neighbors* in (5) learns which neighbors are more relevant and weighs their contributions accordingly. This builds on the long line of successes of attention mechanisms in summarizing long sentences or large images, by capturing which word or part-of-image is most relevant (Xu et al., 2015; Graves et al., 2014; Bahdanau et al., 2014). Particularly, we use the attention formulation similar to the one used in Graves et al. (2014).[2] It only has one parameter and we found this is important for successfully training the model when the number of labels is small as in our semi-supervised learning setting.

## 4.1 AGNN MODEL

We start with a word-embedding layer that maps a bag-of-words representation of a document into an averaged word embedding, and the word embedding $W^{(0)} \in \mathbb{R}^{d_x \times d_h}$ is to be trained as a part of the model:

$$H^{(1)} \quad = \quad \mathrm{ReLU}(XW^{(0)}) \, . \tag{4}$$

This is followed by layers of attention-guided propagation layers parameterized by $\beta^{(t)} \in \mathbb{R}$ at each layer,

$$H^{(t+1)} \quad = \quad P^{(t)} H^{(t)} \, , \tag{5}$$

where the propagation matrix $P^{(t)} \in \mathbb{R}^{n \times n}$ is also a function of the input states $H^{(t)}$ and is zero for absent edges such that the output row-vector of node $i$ is

$$H_i^{(t+1)} \quad = \quad \sum_{j \in N(i) \cup \{i\}} P_{ij}^{(t)} H_j^{(t)} \, ,$$

with $P_i^{(t)} = \mathrm{softmax}\left( [\, \beta^{(t)} \cos(H_i^{(t)}, H_j^{(t)}) \,]_{j \in N(i) \cup \{i\}} \right)$ and $\cos(x, y) = x^T y / \|x\|\|y\|$ with the $L_2$ norm $\|x\|$, for $t \in \{1, \ldots, \ell\}$ and an integer $\ell$. Here $\ell$ is the number of propagation layers. Note that the new propagation above is dynamic; propagation changes over the layers with differing $\beta^{(t)}$ and also the hidden states. It is also adaptive; it learns to weight more relevant neighbors higher. We add the self-loop in the propagation to ensure that the features and the hidden states of the node itself are not lost in the propagation process. The output layer has a weight $W^{(1)} \in \mathbb{R}^{d_h \times d_y}$:

$$Z \quad = \quad f(X, A) \quad = \quad \mathrm{softmax}\left( H^{(\ell+1)} W^{(1)} \right) \, . \tag{6}$$

The weights $W^{(0)}$, $W^{(1)}$, and $\beta^{(t)}$'s are trained on a cross entropy loss in (2). To ease the notations, we have assumed that the input feature vectors to the first and last layers are augmented with a scalar constant of one, so that the standard bias term can be included in the parameters $W^{(0)}$ and $W^{(1)}$.

The softmax function at attention ensures that the propagation layer $P^{(t)}$ row-sums to one. The *attention from node $j$ to node $i$* is

$$P_{ij}^{(t)} \quad = \quad (1/C) e^{\beta^{(t)} \cos(H_i^{(t)}, H_j^{(t)})} \, , \tag{7}$$

with $C = \sum_{j \in N(i) \cup \{i\}} e^{\beta^{(t)} \cos(H_i^{(t)}, H_j^{(t)})}$ which captures how relevant $j$ is to $i$, as measured by the cosine of the angle between the corresponding hidden states. We show how we can interpret the attentions in Section 5.2 and show that the attention selects neighbors with the same class to be more relevant. On the standard benchmark datasets on citation networks, we show in Section 5 that this architecture achieves the best performance in Table 2.

Here we note that independently from this work attention over sets has been proposed as "neighborhood attention" (Duan et al., 2017; Hoshen, 2017) for a different application. The main difference of AGNN with respect to these work is the fact that in AGNN attention is computed over a neighborhood of a node on a graph, whereas in these work attention over set of all entities is used to construct a "soft neighborhood".

---

[2]We have experimented other types of more complex attention formulations but we found the training becomes much less stable especially when the number of labeled nodes is small.

## 5 EXPERIMENTS ON BENCHMARK CITATION NETWORKS

On standard benchmark datasets of three citation networks, we test our proposed AGNN model on semi-supervised learning tasks. We test on a fixed split of labeled/validation/test sets from Yang et al. (2016) and compare against baseline methods in Table 2. We also test it on random splits of the same sizes in Table 3, and random splits with larger number of labeled nodes in Table 4.

**Benchmark Datasets.** A citation network dataset consists of documents as nodes and citation links as directed edges. Each node has a human annotated topic from a finite set of classes and a feature vector. We consider three datasets[3]. For CiteSeer and Cora datasets, the feature vector has binary entries indicating the presence/absence of the corresponding word from a dictionary. For PubMed dataset, the feature vector has real-values entries indicating Term Frequency-Inverse Document Frequency (TF-IDF) of the corresponding word from a dictionary. Although the networks are directed, we use undirected versions of the graphs for all experiments, as is common in all baseline approaches.

| Dataset | Nodes | Edges | Classes | Features | Labeled nodes | | | |
|---|---|---|---|---|---|---|---|---|
| | | | | | Table 2 | Table 3 | Table 4 | |
| CiteSeer | 3,327 | 4,732 | 6 | 3,703 | 120 | 120 | 2,218 | 2,994 |
| Cora | 2,708 | 5,429 | 7 | 1,433 | 140 | 140 | 1,805 | 2,437 |
| PubMed | 19,717 | 44,328 | 3 | 500 | 60 | 60 | 13,145 | 17,745 |

Table 1: Citation Network Dataset

**Experimental setup.** The accuracy of the baseline methods are all taken from existing literature. If a baseline result is not reported in the existing literature, we intentionally left those cells empty in the table for fairness, as opposed to running those experiments ourselves on untuned hyperparameters. We train and test only the two models we propose: GLN for comparisons and our proposed AGNN model. We do not use the validation set labels in training, but use them for optimizing hyper-parameters like dropout rate, learning rate, and $L_2$-regularization factor. For AGNN, we use a fixed number of $d_h = 16$ units in the hidden layers and use 4 propagation layers ($\ell = 4$) for CiteSeer and Pubmed and 3 propagation layers ($\ell = 3$) for Cora as defined in (7). For GLN, we use 2 propagation layers as defined in (1). We row-normalize the input feature vectors, as is standard in the literature. The tables below show the average accuracy with the standard error over 100 training instances with random weight initializations. We implement our model on TensorFlow (Abadi et al., 2016), and the computational complexity of evaluating AGNN is $O(\ell d_h |E| + d_x d_h n)$. Detailed desription of the experiments is provided in Appendix B.

### 5.1 PREDICTION ACCURACY

**Fixed data splits.** In this first experiment, we use the fixed data splits from Yang et al. (2016) as they are the standard benchmark data splits in literature. All experiments are run on the same fixed split of 20 labeled nodes for each class, 500 nodes for validation, 1,000 nodes for test, and the rest of nodes as unlabeled data. Perhaps surprisingly, the linear classifier GLN we proposed in (3) achieves performance comparable to or exceeding the state-of-the-art performance of GCN. This leads to our novel attention-based model AGNN defined in (6), which achieves the best accuracy on all datasets with a gap larger than the standard error. The classification accuracy of all the baseline methods are collected from (Yang et al., 2016; Kipf & Welling, 2016; Such et al., 2017; Monti et al., 2016; Buchnik & Cohen, 2017; Verma et al., 2017).

In semi-supervised learning on graphs, it is critical to utilize both the structure of the graph and the node features. Methods not using all the given data achieve performance far from the state-of-the-art. Supervised methods–Single and Multi-layer Perceptrons–only use the labeled examples $(Y_L, X_L)$. Semi-supervised methods, e.g. T-SVM, only use the labeled and unlabeled examples $Y_L$, and $X$. Skip-gram based approaches, such as DeepWalk, ignores the node features $X$ and only use the labels $Y_L$ and the graph $G$.

---

[3] https://linqs.soe.ucsc.edu/node/236

| Input | Method | CiteSeer | Cora | PubMed |
|---|---|---|---|---|
| $Y_L, X_L$ | Singlelayer Perceptron | 57.2 | 57.4 | 69.8 |
| | Multilayer Perceptron | 64.0 | 57.5 | 71.4 |
| $Y_L, X$ | T-SVM (Joachims, 1999) | 64.0 | 57.5 | 62.2 |
| $Y_L, G$ | DeepWalk (Perozzi et al., 2014) | 43.2 | 67.2 | 65.3 |
| | node2vec (Grover & Leskovec, 2016) | 54.7 | 74.9 | 75.3 |
| | LP (Zhu et al., 2003) | 45.3 | 68.0 | 63.0 |
| | ICA (Lu & Getoor, 2003) | 69.1 | 75.1 | 73.9 |
| | ManiReg (Belkin et al., 2006) | 60.1 | 59.5 | 70.7 |
| | SemiEmb (Weston et al., 2012) | 59.6 | 59.0 | 71.1 |
| | DCNN (Atwood & Towsley, 2016) | | 76.8 | 73.0 |
| | Planetoid (Yang et al., 2016) | 64.7 | 75.7 | 77.2 |
| $Y_L, X, G$ | MoNet (Monti et al., 2016) | | 81.7 | 78.8 |
| | Graph-CNN (Such et al., 2017) | | 76.3 | |
| | DynamicFilter (Verma et al., 2017) | | 81.6 | 79.0 |
| | Bootstrap (Buchnik & Cohen, 2017) | 53.6 | 78.4 | 78.8 |
| | GCN (Kipf & Welling, 2016) | 70.3 | 81.5 | 79.0 |
| | GLN | 70.9±.05 | 81.2±.05 | 78.9±.05 |
| | AGNN (this paper) | **71.7**±.08 | **82.6**±.09 | **79.9**±.07 |

Table 2: Classification accuracy with a fixed split of data from (Yang et al., 2016).

A breakthrough result of Planetoid by Yang et al. (2016) significantly improved upon the existing skip-gram based method of DeepWalk and node2vec and the Laplacian regularized methods of ManiReg and SemiEmb. Kipf & Welling (2016) was the first to apply a graph neural network to citation datasets, and achieved the state-of-the-art performance with GCN. Other variations of graph neural networks immediately followed, achieving comparable performance with MoNet, Graph-CNN, and DynamicFilter. Bootstrap uses a Laplacian regularized approach of (Zhou et al., 2004a) as a sub-routine with bootstrapping to feed high-margin predictions as seeds.

**Random splits.** Next, following the setting of Buchnik & Cohen (2017), we run experiments keeping the same size in labeled, validation, and test sets as in Table 2, but now selecting those nodes uniformly at random. This, along with the fact that different topics have different number of nodes in it, means that the labels might not be spread evenly across the topics. For 20 such randomly drawn dataset splits, average accuracy is shown in Table 3 with the standard error. As we do not force equal number of labeled data for each class, we observe that the performance degrades for all methods compared to Table 2, except for DeepWalk. AGNN achieves the best performance consistently. Here, we note that Kipf & Welling (2016) does a similar but different experiment using GCN, where random labeled nodes are evenly spread across topics so that each topic has exactly 20 labeled examples. As this difference in sampling might affect the accuracy, we do not report those results in this table.

| Method | CiteSeer | Cora | PubMed |
|---|---|---|---|
| DeepWalk (Perozzi et al., 2014) | 47.2 | 70.2 | 72.0 |
| node2vec (Grover & Leskovec, 2016) | 47.3 | 72.9 | 72.4 |
| Bootstrap (Buchnik & Cohen, 2017) | 50.3 | 78.2 | 75.6 |
| GLN | 68.4±0.45 | 80.0±0.43 | 77.7±0.63 |
| AGNN (this paper) | **69.8**±0.35 | **81.0**±0.34 | **78.0**±0.46 |

Table 3: Classification accuracy with random splits of the data.

**Larger training set.** Following the setting of Such et al. (2017), we run experiments with larger number of labeled data on Cora dataset. We perform $k$-fold cross validation experiments for $k = 3$ and 10, by uniformly and randomly dividing the nodes into $k$ equal sized partitions and then performing $k$ runs of training by masking the labels of each of the $k$ partitions followed by validation on the masked nodes. Finally the average validation accuracy across $k$ runs is reported. We run 10 trials of this experiment and reports the mean and standard error of the average $k$-fold validation

accuracy. Compared to Table 2, the performance increases with the size of the training set, and AGNN consistently outperforms the current state-of-the-art architecture for this experiment.

| Method | 3-fold Split | 10-fold split |
|---|---|---|
| Graph-CNN (Such et al., 2017) | 87.55±1.38 | 89.18±1.96 |
| GLN | 87.98±0.08 | 88.24±0.07 |
| AGNN (this paper) | **89.07**±0.08 | **89.60**±0.09 |

Table 4: Classification accuracy with larger sets of labelled nodes.

## 5.2 QUALITATIVE ANALYSIS

One useful aspect of incorporating attention into a model is that it provides some form of interpretation capability (Bahdanau et al., 2014). The learned $P_{ij}^{(t)}$'s in Eq. (7) represent the *attention* from node $j$ to node $i$, and provide insights on how relevant node $j$ is in classifying node $i$. In Figure 1, we provide statistics of this attention over all adjacent pairs of nodes for Cora and CiteSeer datasets. We refer to Figure 3 for similar statistics on PubMed. In Figure 1, we show average attention from a node in topic $c_2$ (column) to a node in topic $c_1$ (row), which we call the *relevance* from $c_2$ to $c_1$ and is defined as

$$\text{Relevance}(c_2 \to c_1) \quad = \quad \frac{1}{|S_{c_1,c_2}|} \sum_{(i,j) \in S_{c_1,c_2}} R(j \to i) \,, \tag{8}$$

for edge-wise relevance score defined as

$$R(j \to i) \quad = \quad \left( P_{ij}^{(t)} - \frac{1}{|N(i)| + 1} \right) \Big/ \frac{1}{|N(i)| + 1} \,, \tag{9}$$

where $|N(i)|$ is the degree of node $i$, and $S_{c_1,c_2} = \{(i,j) \in E^s \text{ and } Y_i = c_1, Y_j = c_2\}$ where $E^s = E \cup \{(i,i) \text{ for } i \in V\}$ is the edge set augmented with self-loops to include all the attentions learned. If we are not using any attention, then the typical propagation will be uniform $P_{ij} = 1/(|N(i)|+1)$, in which case the above normalized attention is zero. We are measuring for each edge the variation of attention $P_{ij}$ from uniform $1/(|N(i)| + 1)$ as a *multiplicative error*, normalized by $1/(|N(i)| + 1)$. We believe this is the right normalization, as attention should be measure in relative strength to others in the same neighborhood, and not in absolute additive differences. We are measuring this multiplicative variation of the attention, averaged over the ordered pairs of classes.

Figure 1 shows the relevance score for CiteSeer and Cora datasets. (PubMed is shown in Appendix A.) For both datasets, the diagonal entries are dominant indicating that the attention is learning to put more weight to those in the same class. A higher value of Relevance($c_2 \to c_1$) indicates that, on average, a node in topic $c_1$ pays more attention to a neighbor in topic $c_2$ than neighbors from other topics. For CiteSeer dataset (Figure 1 left), we are showing the average attention at the first propagation layer, $P^{(t=1)}$, for illustration. In the off-diagonals, the most influential relations are HCI→Agents, Agents→ML, Agents→HCI, and ML→Agents, and the least influential relations are AI→IR and DB→ML. Note that these are papers in computer science from late 90s to early 2000s. For Cora dataset (Figure 1 right), we are showing the relevance score of the second propagation layer, $P^{(t=2)}$, for illustration. In the off-diagonals, the most influential relations are CB→PM, PM→CB, Rule→PM, and PM→Rule, and the least influential relations are GA→PM and PM→RL. This dataset has papers in computer science from the 90s. We note that these relations are estimated solely based on the available datasets for that period of time and might not accurately reflect the relations for the entire academic fields. We also consider these relations as a static property in this analysis. If we have a larger corpus over longer period of time, it is possible to learn the influence conditioned on the period and visualize how these relations change.

Next, we analyze the edges with high and low relevance scores. We remove the self-loops and then sort the edges according to the relevance score defined in Eq. (9). We take the top 100 and bottom 100 edges and with respect to their relevance scores, and report the fraction of the edges which are connecting nodes from the same class. Table 5 shows the result on the benchmark datasets for the relevance scores calculated using the last propagation layer. This suggests that our architecture learns to put higher attention between nodes of the same class.

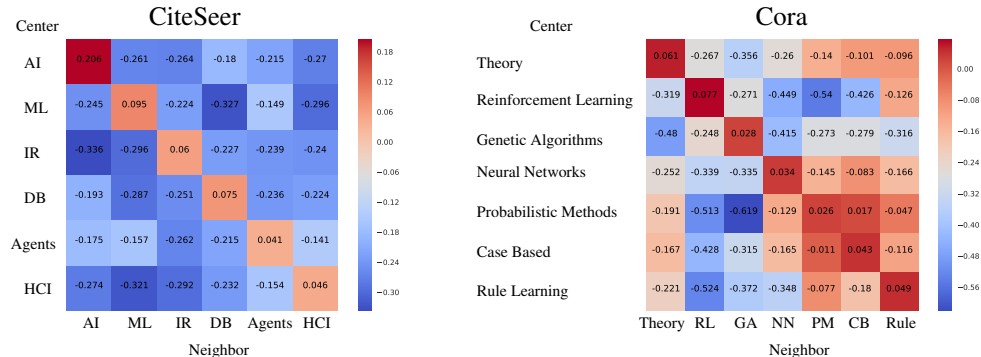

Figure 1: Relevance score in Eq. (8) from a neighbor node with column-class to a center node in row-class. For example the average normalized attention to Agents from HCI is $= -0.141$ , largest off-diagonal entry in CiteSeer. The average attention to Probabilistic Methods (PM) from Case Based (CB) is $0.017$, largest off-diagonal entry in Cora.

|            | CiteSeer | Cora | PubMed |
|------------|----------|------|--------|
| Top 100    | 0.69     | 0.64 | 0.69   |
| Bottom 100 | 0.34     | 0.09 | 0.13   |

Table 5: Fraction of edges from top 100 most relevant edges and bottom 100 least relevant edges which are connecting two distinct nodes from the same class.

Finally, we analyze those nodes in the test sets that were mistaken by GCN but correctly classified by AGNN, and show how our attention mechanism weighted the contribution of its local neighborhood, and show three illustrative examples in Figure 2. More examples of this local attention network (including the legends for the color coding of the topics) are provided in Appendix A. We show a entire 2-hop neighborhood of a target node (marked by a thick outline) from the test set of the fixed data splits of Citeseer, Cora, or Pubmed. The colors denote the true classes of the nodes (including the target) in the target's neighborhood, some of which are unknown to the models at the training time. The radius of a node $j$ is proportional to the attention to the target node $i$ aggregated over all the layers, i.e. $\left(P^{(t=4)}P^{(t=3)}P^{(t=2)}P^{(t=1)}\right)_{ij}$ for CiteSeer. The size of the target node reflects its self-attention defined in a similar way. The first example on the left is node 8434 from PubMed. AGNN correctly classifies the target node as light blue, whereas GCN mistakes it for yellow, possibly because it is connected to more yellow nodes. Not only has the attention mechanism learned to put more weight to its light blue 1-hop neighbor, but put equally heavy weights to a path of light blue neighbors some of which are not immediately connected to the target node. The second example in the middle is node 1580 from PubMed. AGNN correctly classifies it as yellow, whereas GCN mistakes it for a red, possibly because it only has two neighbors. Not only has the attention mechanism learned to put more weight to the yellow neighbor, but it has weighted the yellow neighbor (who is connected to many yellow nodes and perhaps has more reliable hidden states representing the true yellow class) even more than itself.

The last example on the right is node 1512 from CiteSeer. AGNN correctly classifies it as light blue, whereas GCN mistakes it for a white. This is a special example as those two nodes are completely isolated. Due to the static and non-adaptive propagation of GCN, it ends up giving the same prediction for such isolated pairs. If the pair has two different true classes, then it always fails on at least on one of them (in this case the light blue node). However, AGNN is more flexible in adapting to such graph topology and puts more weight to the target node itself, correctly classifying both.

## 6 CONCLUSIONS

In this paper, we present an attention-based graph neural network model for semi-supervised classification on a graph. We demonstrate that our method consistently outperforms competing methods on the standard benchmark citation network datasets. We also show that the learned attention also provides interesting insights on how neighbors influence each other. In training, we have tried more

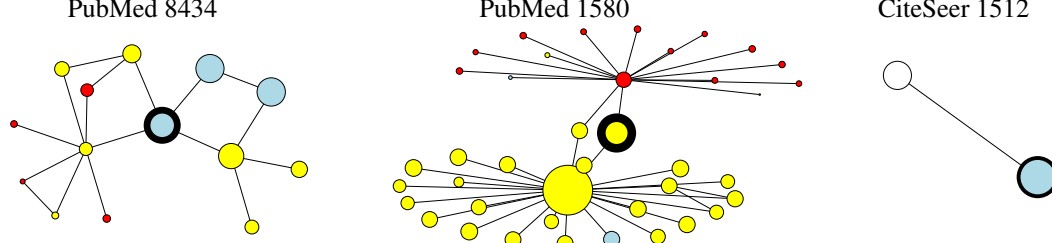

Figure 2: We show three selected target nodes in the test set that are mistaken by GCN but correctly classified by AGNN. We denote this target node by the node with a thick outline (node 8434 from PubMed on the left, node 1580 from PubMed in the middle, and node 1512 from CiteSeer on the right). We show the strength of attention from a node in the 2-hop neighborhood to the target node by the size of the corresponding node. Colors represent the hidden true classes (nodes with the same color belong to the same topic). None of the nodes in the figure was in the training set, hence none of the colors were revealed. Still, we observe that AGNN has managed to put more attention to those nodes in the same (hidden) classes, allowing the trained model to find the correct labels.

complex attention models. However, due to the increased model complexity the training was not stable and does not give higher accuracy. We believe that for semi-supervised setting with such a limited number of labeled examples, reducing model complexity is important. Note that we are able to train deeper (4-layers) models compared to a shallower (2-layers) model of GCN, in part due to the fact that we remove the non-linear layers and reduce the model complexity significantly. In comparison, deeper GCN models are known to be unstable and do not give the performance of shallower GCNs (Kipf & Welling, 2016).

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

APPENDIX

## A    ADDITIONAL EXPERIMENTS ON INTERPRETABILITY OF ATTENTION

PubMed dataset has only 3 classes, and the relevance score is shown in the figure below.

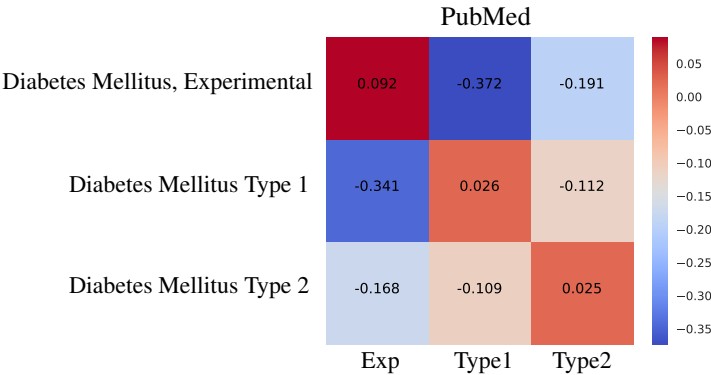

Figure 3: Average attention in Eq. (8) from a column class to a row class

We show examples of 2-hop local neighborhood of nodes that are mistaken by GCN but correctly classified by AGNN.

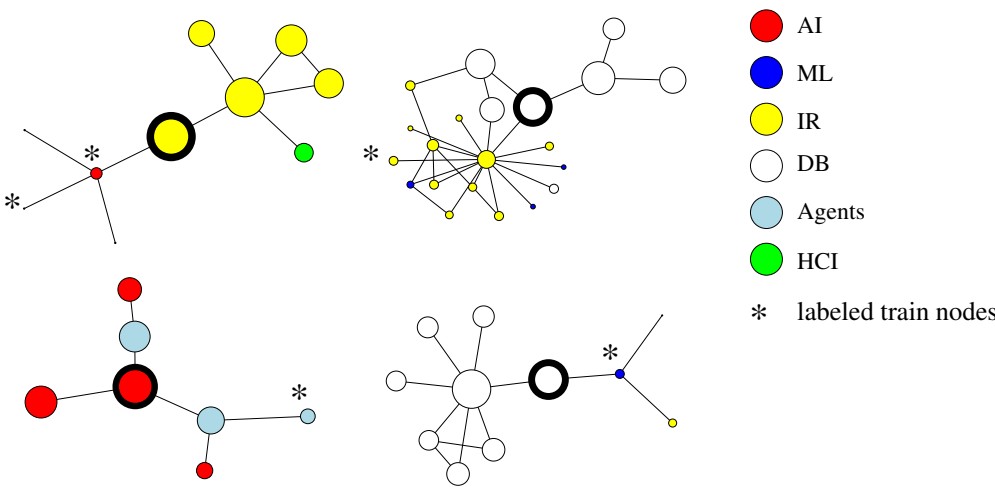

Figure 4: Examples from CiteSeer dataset of attention strength in the local neighborhood of a target node (in thick outline) from the test set that is mistaken by GCN but correctly classified by AGNN. Colors are true classes and node sizes are proportional to the attention strength from a neighbor to the target node. Labeled nodes from training set are marked with '*'.

## B    EXPERIMENT AND ARCHITECT DETAILS

In this section we will list all the choices made in training and tuning of hyper-parameters. The parameters are chosen as to maximize the validation. All the models use Adam optimization algorithm with full-batchs, as standard in other works on GNNs (Kipf & Welling, 2016; Such et al., 2017). We also a weight decay term to the objective function for all the learnable weights. We add dropout to the first and last layers of all models.

In Table 6. we show the hyper-parameters used in training the AGNN models for various settings and datasets. For Cora dataset the architecture consist of $\ell = 3$ propagation layers, but the first prop-

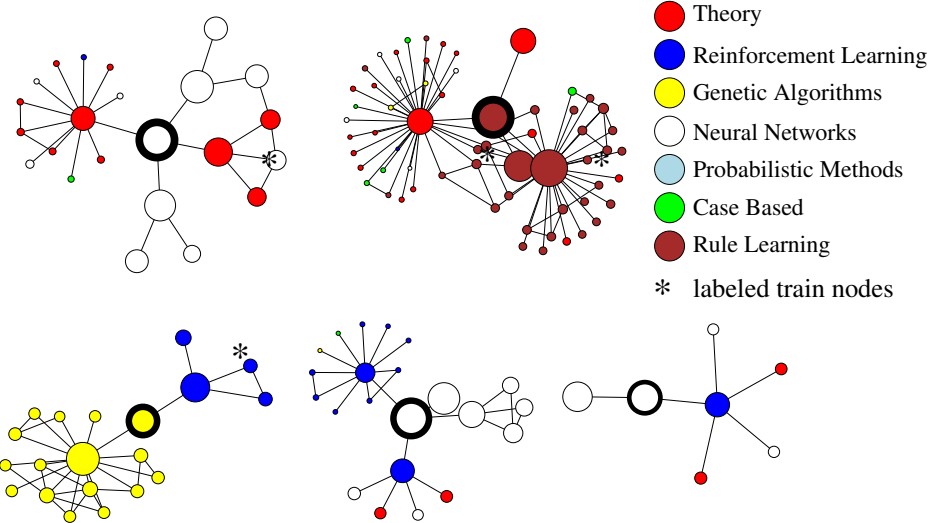

Figure 5: Examples from Cora dataset of attention strength in the local neighborhood of a target node (in thick outline) from the test set that is mistaken by GCN but correctly classified by AGNN. Colors are true classes and node sizes are proportional to the attention strength from a neighbor to the target node. Labeled nodes from training set are marked with '*'.

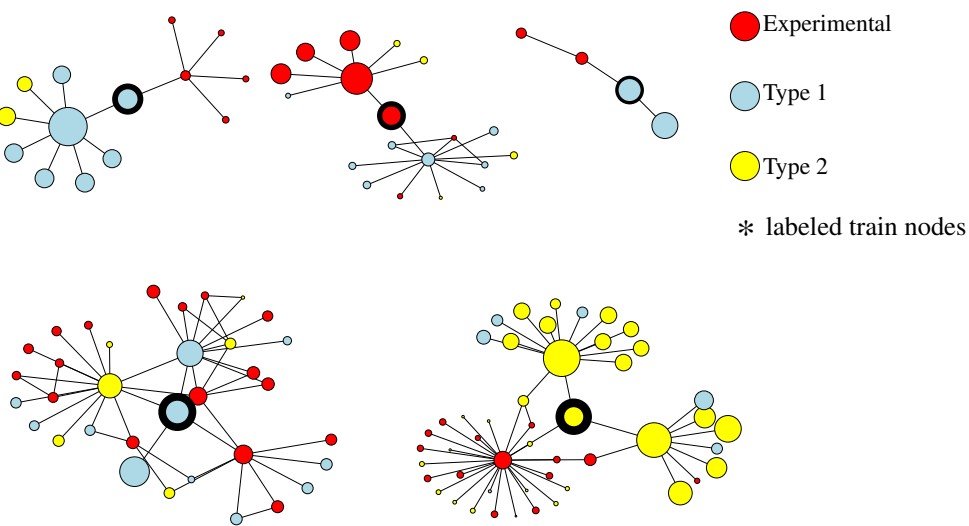

Figure 6: Examples from Pubmed dataset of attention strength in the local neighborhood of a target node (in thick outline) from the test set that is mistaken by GCN but correctly classified by AGNN. Colors are true classes and node sizes are proportional to the attention strength from a neighbor to the target node. Labeled nodes from training set are marked with '*'.

agation layer $P^{(t=1)}$ is non-trainable and the variable $\beta^{(t=1)}$ value is fixed at zero. While training these AGNN models we maintain the validation accuracy for each iteration and finally choose the trained model parameters from the iteration where average validation accuracy of previous 4 epochs is maximized. For the $k$-fold cross validation setting we take the epoch with maximum validation accuracy.

| Setting | Dataset | Propagation layers ($\ell$) | Hidden state dimension ($d_h$) | Learning Rate | Weight Decay | Dropout | Epochs |
|---|---|---|---|---|---|---|---|
| Fixed Split | CiteSeer | 4 | 16 | 0.005 | 0.0005 | 0.5 | 1000 |
| Fixed Split | Cora | 3 | 16 | 0.01 | 0.0005 | 0.5 | 1000 |
| Fixed Split | PubMed | 4 | 16 | 0.008 | 0.001 | 0.5 | 400 |
| Rand. Split | CiteSeer | 4 | 16 | 0.01 | 0.0005 | 0.5 | 1000 |
| Rand. Split | Cora | 3 | 16 | 0.01 | 0.0005 | 0.5 | 1000 |
| Rand. Split | PubMed | 4 | 16 | 0.008 | 0.001 | 0.5 | 1000 |
| Cross Val. | Cora | 3 | 16 | 0.04 | 0.0005 | 0.25 | 500 |

Table 6: Hyper-parameters for AGNN model

For the Graph Linear Network (GLN) as define in (3), we use the same hyper-parameters as GCN (Kipf & Welling, 2016) for all the experimental settings: hidden dimension of 16, learning rate of 0.01, weight decay of $5 \times 10^{-4}$, dropout of 0.5, 200 epochs and early stopping criteria with a window size of 10.

## C  PERFORMANCE VERSUS NUMBER OF PROPAGATION LAYERS

In this section we provide experimental results justifying the choice of number of propagation layers for each dataset. We use the same data split and hyper-parameters (except number of propagation layer) as in the fixed data splits setting in Section 5.1. Similar to other settings the first propagation layer for Cora dataset is non-trainable with $\beta^{(t=1)} = 0$. Tables 7 gives the average (over 10 trials) testing accuracy respectively for various choices of number of propagation layers. We note that different datasets require different number of propagation layers for best performance.

| Propagation Layers ($\ell$) | CiteSeer | Cora | PubMed |
|---|---|---|---|
| 1 | 69.08 | 80.52 | 78.31 |
| 2 | 70.83 | 83.07 | 79.66 |
| 3 | 71.06 | 82.62 | 79.56 |
| 4 | 71.70 | 82.24 | 80.10 |
| 5 | 71.37 | 82.07 | 80.13 |

Table 7: Average testing error of AGNN for different number of Propagation Layers.

## D  PERFORMANCE OF GCN ON OTHER DATASET SPLITS.

| Method | CiteSeer | Cora | PubMed |
|---|---|---|---|
| DeepWalk (Perozzi et al., 2014) | 47.2 | 70.2 | 72.0 |
| node2vec (Grover & Leskovec, 2016) | 47.3 | 72.9 | 72.4 |
| Bootstrap (Buchnik & Cohen, 2017) | 50.3 | 78.2 | 75.6 |
| GCN | 66.9±0.50 | 79.2±0.46 | 77.5±0.61 |
| GLN | 68.4±0.45 | 80.0±0.43 | 77.7±0.63 |
| AGNN (this paper) | **69.8**±0.35 | **81.0**±0.34 | **78.0**±0.46 |

Table 8: Classification accuracy with random splits of the data.

Here we provide the performance of GCN on **random splits** and **larger training set** dataset splits from Section 5.1. We relegate these tables to the appendix since they were not present in Kipf & Welling (2016). We conducted the experiments with the same hyper-parameters as chosen by Kipf & Welling (2016) for the **fixed split**. In Tables 8 and 9 we provide average testing accuracy and standard error over 20 and 10 runs on **random splits** and **larger training set** respectively.

| Method | 3-fold Split | 10-fold split |
|---|---|---|
| Graph-CNN (Such et al., 2017) | 87.55±1.38 | 89.18±1.96 |
| GCN | 87.23±0.21 | 87.78±0.04 |
| GLN | 87.98±0.08 | 88.24±0.07 |
| AGNN (this paper) | **89.07**±0.08 | **89.60**±0.09 |

Table 9: Classification accuracy with larger sets of labelled nodes.

