# OpenReview forum: "Attention-based Graph Neural Network for Semi-supervised Learning"
_ICLR.cc/2018/Conference — Reject_

### Official Review · AnonReviewer2 · 2017-11-25
**idea would be reasonable and constains interesting insight**

**Rating:** 6
**Confidence:** 2

**Review:**

The paper proposes graph-based neural network in which weights from neighboring nodes are adaptively determined. The paper shows importance of propagation layer while showing the non-linear layer does not have significant effect. Further the proposed method also provides class relation based on the edge-wise relevance.

The paper is easy to follow and the idea would be reasonable.

Importance of the propagation layer than the non-linear layer is interesting, and I think it is worth showing.

Variance of results of AGNN is comparable or even smaller than GLN. This is a bit surprising because AGNN would be more complicated computation than GLN. Is there any good explanation of this low variance of AGNN?

Interpretation of Figure 2 is not clear. All colored nodes except for the thick circle are labeled node? I couldn't judge those predictions are appropriate or not.

---

> ### Author Response · Authors · 2018-01-04
> **Response to AnonReviewer2: Improved Figure 2 for qualitative analysis of Attention**
>
> Thank you for your time, review and valuable comments.
>
> 1. Regarding the similar variance of results of AGNN and GLN: In Table 2 of the original version we don’t report the variance or standard-deviation of accuracies of the trials, but we report (as mentioned in paragraph 1 on page 3 of original version) standard-error which defined as standard-deviation/square-root(number of trials) (https://en.wikipedia.org/wiki/Standard_error).  That being said, when the training data is fixed (as is the case for Table 2), the variance of GLN is smaller than that of AGNN as predicted by the reviewer, as the only source of randomness is the initialization of the neural network weights. On the other hand, when the training data is chosen randomly (As is the case for Tables 3 and 4), there are two sources of randomness and the variance of GLN and AGNN are harder to predict and compare. We could not predict how different choices of the training data affects the accuracy, and it can happen that GLN has larger variance than AGNN.
>
> 2. Regarding Figure 2.: We apologize for the lack of clarity in its caption. The thick nodes are from the test set whose labels are not known to the model at training time. For clarification, we have now added `*’ (asterisk) to mark nodes from the training set whose labels were revealed to the model during training (e.g. Figure 4). Coincidentally none of the neighborhood in Figure 2 have any nodes from the training set.

---

### Official Review · AnonReviewer3 · 2017-11-29
**Interesting paper with nice findings. The originality is however relatively limited in a field where many recent papers have been proposed, and the experiments need to be completed.**

**Rating:** 7
**Confidence:** 4

**Review:**

The paper proposes a semi supervised learning algorithm for graph node classification. The Algorithm is inspired from Graph Neural Networks and more precisely graph convolutional NNs recently proposed by ref (Kipf et al  2016)) in the paper.  These NNs alternate 2 types of layers:  non linear projection and diffusion, the latter incorporates the graph relational information by constraining neighbor nodes to have close representations according to some “graph metrics”. The authors propose a model with simplified projection layers and more sophisticated diffusion ones, incorporating a simple attention mechanism. Experiments are performed on citation textual datasets. Comparisons with published results on the same datasets are presented.

The paper is clear and develops interesting ideas relevant to semi-supervised graph node classification. One finding is that simple models perform as well as more complex ones in this setting where labeled data is scarce. Another one is the importance of integrating relational information for classifying nodes when it is available. The attention mechanism itself is extremely simple, and learns one parameter per diffusion layers. One parameter weights correlations between node embeddings in a diffusion layer. I understand that you tried more complex attention mechanisms, but the one finally selected is barely an attention mechanism and rather a simple “importance” weight. This is not a criticism, but this makes the title somewhat misleading. The experiments show that the proposed model is state of the art for graph node classification. The performance is on par with some other recent models according to table 2. The other tests are also interesting, but the comparison could have been extended to other models e.g. GCN.
You advocate the role of the diffusion layers, and in the experiments you stack 3 to 4 such layers. It would be interesting to have indications on the compromise performance/ number of diffusion layers and on the evolution of these performances when adding such layers.
The bibliography on semi-supervised learning in graphs for classification is light and should be enhanced.
Overall this is an interesting paper with nice findings. The originality is however relatively limited in a field where many recent papers have been proposed, and the experiments need to be completed.

---

> ### Author Response · Authors · 2018-01-04
> **Response to AnonReviewer3: Compared AGNN with different number of layers and added experiments with GCN**
>
> Thank you for reviewing our paper and pointing out missed experiments and inconsistencies.
>
> 1. Attention mechanism:  It is true as the reviewer pointed out that our attention mechanism is very simple. We settled on this choice after training/testing several attention mechanisms, most of which are more complex than the one we propose. The proposed simple attention mechanism gave the best performance, among those we tried. We believe this is due to the fact that complex attention mechanisms are harder to train as there are more parameters to learn.
>
> 2. GCN on other training sets: The reason we do not report GCN performance in tables 2 and 3 is that we made it our rule not to run other researcher’s algorithms ourselves, at the fear of not doing justice in the hyperparameters we need to choose.   However, given the interest in the numerical comparisons, as the reviewer pointed out, in the revised version, we run these experiments  and reported the performance of GCN in the appendix D (as it might give the wrong impression that those results are performed by the authors of GCN, if we put it in the table in the main text).
>
> 3. Choice of number of diffusion layers: Thanks for pointing this out. We have added a table in the appendix C which contains testing accuracies of AGNN model with different number of diffusion layers.
>
> 4. Regarding bibliography: We have expanded the bibliography on semi-supervised learning using graphs. Please see the section 2 in the revised manuscript.

---

### Official Review · AnonReviewer4 · 2017-12-01
**few questions**

**Rating:** 6
**Confidence:** 3

**Review:**

SUMMARY.

The paper presents an extension of graph convolutional networks.
Graph convolutional networks are able to model nodes in a graph taking into consideration the structure of the graph.
The authors propose two extensions of GCNs, they first remove intermediate non-linearities from the GCN computation, and then they add an attention mechanism in the aggregation layer, in order to weight the contribution of neighboring nodes in the creation of the new node representation.
Interestingly, the proposed linear model obtains results that are on-par with the state-of-the-art model, and the linear model with attention outperforms the state-of-the-art models on several standard benchmarks.


----------

OVERALL JUDGMENT
The paper is, for the most part, clear, although some improvement on the presentation would be good (see below).
An important issue the authors should address is the notation consistency, the indexes i and j are used for defining nodes and labels, please use another index for labels.
It is very interesting that stripping standard GCN out of nonlinearities gives pretty much the same results, I would appreciate if the authors could give some insights of why this is the case.
It seems to me that an important experiment is missing here, have the authors tried to apply the attention model with the standard GCN?
I like the idea of using a very minimal attention mechanism. The similarity function used for the attention (cosine) is symmetric, this means that if two nodes are connected in both directions, they will be equally important for each other. But intuitively this is not true in general. It would be interesting if the authors could elaborate a bit more on the choice of the similarity function.


----------

DETAILED COMMENTS
Page 2. I do not understand the point of so many details on Graph Laplacian Regularization.
Page 2. The use of the term 'skip-grams' is somewhat odd, it is not clear what the authors mean with that.
Page 3. 'the natural random walk' ???
Bottom of page 4. When the authors introduce the attention based network also introduce the input/embedding layer, I believe there is a better place to do so instead of that together with the most important contribution of the paper.

---

> ### Author Response · Authors · 2018-01-04
> **Response to AnonReviewer4**
>
> We are thankful for your review and insightful comments.
>
> 1. Confusing notation is corrected: In the revised version $c$ indexes a label.
>
> 2. Why GLN works:  For semi-supervised learning, we believe that the primary gain of using graph neural network comes from the “Averaging” effect. Similar to denoising pixels in images, by averaging neighbors features, we get a denoised version of current nodes’ features. This gives significant gain over those estimations without denoising (such as Mulit-Layer Perceptron in Table 2). This, we believe, is why GLN is already achieving the state-of-the-art performance. The focus of this paper is how to get the next remaining gain, which we achieve by proposing asymmetric averaging using “attention”. So far, we did not see any noticeable gain in non-linear activation for semi-supervised learning. However, we believe such non-linearity can be important for other applications, such as graph classification tasks on molecular networks.
>
> 3. Attention in GCN: GCN with attention did not give gain over our AGNN architecture, which is somewhat expected as GCN and GLN have comparable performances, within the error margin of each other. Note that from the architecture complexity perspective AGNN is simpler than GCN with attention, meaning that AGNN might have a better chance explaining the data.
>
> 4. Symmetric attention: Even though the scaled cosine similarity would be symmetric between two connected nodes $i$ and $j$, the attention value itself can be different due to the fact that softmax computations are calculated on different neighborhoods: $N(i)$ and $N(j)$ respectively.
> But we agree that attention mechanism has an element of symmetry and this might be alleviated by using more complex attention mechanism. As the reviewer pointed out, we chose the simple attention mechanism here; we tried various attention mechanisms with varying degrees of complexity, and found the simple attention mechanism to give the best performance. Training complex attention is challenging, and we would like to explore more complex ones in our future work.
>
> Response to detailed comments:
>
> 1. Details on Graph Laplacian Regularization: We added details about Laplacian regularization for completeness of discussion of previous work and because Laplacian regularizations closely related to the propagations layers used in almost all Graph Neural Network papers.
> 2. ‘Skip-grams’: We added some clarification on the use of ‘skip-grams’ in the revised version.
> 3. ‘Natural random walk’ on a graph is random walk where one move from a node to one of its neighbors selected with uniform probability. We have clarified this in the revised version.
> 4. Presentation of the Attention-based Graph Neural Network: Thanks for pointing this out. We have made some changes to the presentation style.

---

> ### Public Comment · ~Xiao-Ming_Wu1 · 2018-01-25
> **Why GCNs or GLNs work**
>
> To add to the discussion, we would like to draw the attention to our recent paper accepted by AAAI-2018 as oral presentation (https://arxiv.org/abs/1801.07606). We have shown in our paper that the convolution layer of GCNs acts as "Laplacian smoothing" on the vertex features, which is the key reason why GCNs work. This may also help explain why the GLN model works just as well as the GCN model.

---

### Public Comment · ~Yedid_Hoshen1 · 2017-11-08
**Related Work**

Nice work!

It would be interesting to compare this work and VAIN (NIPS'17) https://arxiv.org/abs/1706.06122

---

> ### Author Response · Authors · 2017-11-13
> **Thank you for pointing out the missing reference.**
>
> Thank you for your interest in our paper and bringing the VAIN model to our attention.
>
> We see that VAIN uses attention between multiple-agents in a system. We were not aware of this line of literature when we submitted the paper. We will cite this line of work in our final version. Below is a comparison between VAIN and Attention-based-Graph Neural Network.
>
> Main similarities and differences between VAIN and Attention-based Graph Neural Network (AGNN) are as follows:
> 1. Experimental results: AGNN is tested on semi-supervised classification of nodes on a graph where as VAIN is tested on prediction of future state in multi-agent systems.
>
> 2. Side information: AGNN is graph processing neural network, but VAIN model does not take graph as an input as initially proposed (although it could).  In VAIN, it is assumed that every agent can possibly interact with every other agent in the system. Where as in AGNN we have a known graph in which real world first order interaction between two nodes are represented as edges and attention is computed only for these first order interactions. VAIN clubs all the higher order (long range) interactions into a single attention mechanism, where as AGNN computes higher order interactions through multiple hops of first order attention mechanism.

---

### Public Comment · ~Thomas_N._Kipf1 · 2017-11-20
**Linear model and further references on attention-based models**

Very interesting work!

Your insight about using a linear activation function on the hidden layers of a graph neural net looks interesting and indeed simplifies this class of models significantly. Have you been able to verify this architecture on some more challenging tasks, e.g. for molecule classification or the tasks presented in https://arxiv.org/abs/1511.05493, where graph neural networks typically show very strong performance as well?

I am also wondering about your choice of hyper parameters when you compare your linear model to the one in https://arxiv.org/abs/1609.02907 : do you similarly use the same set of hyper parameters for all three datasets (Cora, Citeseer and Pubmed), or do you tune them individually? To make your result stronger, it would be good to tune the baseline (GCN) with the same procedure and include GCN baseline results on all of your experiments - it should be very simple by just running: https://github.com/tkipf/gcn/

As noted by Yedid Hoshen, it would be good if you could refer to some earlier work on graph neural networks with attention mechanisms:

https://arxiv.org/abs/1703.07326 - Introduces "Neighborhood attention"
https://arxiv.org/abs/1706.06383 - Improved version of "Neighborhood attention"
https://arxiv.org/abs/1706.06122 - Attention mechanism in a graph neural net model for multi-agent reinforcement learning (as noted by Yedid Hoshen)

---

> ### Author Response · Authors · 2017-11-21
> **Thank you for the insightful comment.**
>
> We agree that graph classification is another exciting application where graph neural networks are making breakthroughs. There are several key differences in the dataset (from citation networks) and we have not tried the idea of linear architecture for the molecular dataset yet. For example, the edges have attributes. There are straight forward ways to incorporate such information into GNNs, but we have not pursued this direction yet. I do agree the experiments you suggested will both (a) clarify what the gain is in non-linear activation; and (b) give insights on how different datasets (and applications) might require different architectures.
>
> For the linear model, we did not tune the hyper parameters and the same hyper parameters are used as your (Kipf and Welling) original GCN. We made a small change in the stopping criteria  to take the best model in validation error out of all epochs. We did not see any significant change when we use the same stopping criteria as GCN. We will make this explicit during the revision process. Overall, there was no hyperparameter tuning for the linear model, and all the numbers should provide fair comparisons.
>
> Thank you for the references, we will surely include and discuss all the great prior work you pointed out.

---

### Author Response · Authors · 2018-01-04
**Changes made to the paper**

We thank the reviewers and the other commenters for helping us improve our work and its presentation. Taking the reviews and comments to heart we have made several changes which, we believe, greatly improve our paper. We added comparison of performance of AGNN with different number of propagation layers in Appendix C. In the Appendix D, we added experimental results of GCN on both random splits and cross-validation settings. Further, we have expanded the bibliography in the Sections 2 and 4.1. As per the reviews we have also made changes to notations and presentation style in Sections 3 and 4. In Section 5.2 and Appendix A, we corrected the order of class names. We improved the caption and marked training set nodes in Figures 2, 4, 5 and 6. Finally, we made some minor changes in the text.

---

### Decision · Program_Chairs · 2018-01-29
**ICLR 2018 Conference Acceptance Decision**

**Decision:**

Reject

**Comment:**

A version of GCNs of Kipf and Welling is introduced with (1) no non-linearity; (2) a basic form of (softmax) attention over neighbors where the attention scores are computed as the cosine of endpoints' representations (scaled with a single learned scalar). There is a moderate improvement on Citeseer, Cora, Pubmed.

Since the use of gates with GCNs / Graph neural networks is becoming increasingly common (starting perhaps with GGSNNs of Li et al, ICLR 2016)) and using attention in graph neural networks is also not new  (see reviews and comments for references), the novelty is very limited.  In order to make the submission more convincing the authors could: (1) present results on harder datasets; (2)  carefully evaluate against other forms of attention (i.e. previous work).

As it stands, though it is interesting to see that such simple model performs well on the three datasets, I do not see it as an ICLR paper.

Pros:
-- a simple model, achieves results close / on par with state of the art

Cons:
-- limited originality
-- either results on harder datasets or / and evaluation agains other forms of attention (i.e. previous work) are needed